# Alternative Oxidase Alleviates Mitochondrial Oxidative Stress during Limited Nitrate Reduction in *Arabidopsis thaliana*

**DOI:** 10.3390/biom14080989

**Published:** 2024-08-11

**Authors:** Daisuke Otomaru, Natsumi Ooi, Kota Monden, Takamasa Suzuki, Ko Noguchi, Tsuyoshi Nakagawa, Takushi Hachiya

**Affiliations:** 1Department of Molecular and Functional Genomics, Interdisciplinary Center for Science Research, Shimane University, 1060 Nishikawatsu-cho, Matsue 690-8504, Shimane, Japan; n23m803@matsu.shimane-u.ac.jp (D.O.); n23d203@matsu.shimane-u.ac.jp (K.M.); tnakagaw@life.shimane-u.ac.jp (T.N.); 2Division of Biological Science, Nara Institute of Science and Technology, 8916-5 Takayama, Ikoma 630-0192, Nara, Japan; 3College of Bioscience and Biotechnology, Chubu University, 1200 Matsumoto-cho, Kasugai 487-8501, Aichi, Japan; takamasa@tsbio.info; 4Department of Life Sciences, Tokyo University of Pharmacy and Life Sciences, 1432-1 Horinouchi Hachioji, Tokyo 192-0392, Japan; knoguchi@toyaku.ac.jp

**Keywords:** alternative oxidase, mitochondrial dysfunction stimulon, nitrate reduction, oxidative stress

## Abstract

The conversion of nitrate to ammonium, i.e., nitrate reduction, is a major consumer of reductants in plants. Previous studies have reported that the mitochondrial alternative oxidase (AOX) is upregulated under limited nitrate reduction conditions, including no/low nitrate or when ammonium is the sole nitrogen (N) source. Electron transfer from ubiquinone to AOX bypasses the proton-pumping complexes III and IV, thereby consuming reductants efficiently. Thus, upregulated AOX under limited nitrate reduction may dissipate excessive reductants and thereby attenuate oxidative stress. Nevertheless, so far there is no firm evidence for this hypothesis due to the lack of experimental systems to analyze the direct relationship between nitrate reduction and AOX. We therefore developed a novel culturing system for *A. thaliana* that manipulates shoot activities of nitrate reduction and AOX separately without causing N starvation, ammonium toxicity, or lack of nitrate signal. Using shoots processed with this system, we examined genome-wide gene expression and growth to better understand the relationship between AOX and nitrate reduction. The results showed that, only when nitrate reduction was limited, AOX deficiency significantly upregulated genes involved in mitochondrial oxidative stress, reductant shuttles, and non-phosphorylating bypasses of the respiratory chain, and inhibited growth. Thus, we conclude that AOX alleviates mitochondrial oxidative stress and sustains plant growth under limited nitrate reduction.

## 1. Introduction

Plants use nitrate and ammonium as main sources of nitrogen (N). The conversion of nitrate to ammonium, i.e., nitrate reduction, requires eight electrons per nitrate, which accounts for about half of the energy required for protein synthesis from nitrate [1]. Thus, ammonium is an energetically superior N source, but most crops prefer nitrate [2]. In herbaceous plants, nitrate reduction generally occurs in the shoots [3,4,5], and acts as an electron sink for reductants such as NAD(P)H and reduced ferredoxin that are generated in the cytosol and chloroplasts [6]. Hence, when the nitrate supply is limited, excessive reductants may accumulate within illuminated photosynthetic cells. Recent studies have shown that the NAD(P)H/NAD(P)^+^ ratio and the levels of hydrogen peroxide are increased in *A. thaliana* leaves of ammonium-grown plants compared with nitrate-grown plants [6,7,8]. Moreover, nitrate deficiency treatments have been found to increase the levels of superoxide, hydrogen peroxide, and malondialdehyde in tobacco leaves [9]. These findings suggest that a decrease in nitrate reduction could lead to oxidative stress in plants.

It is widely accepted that nitrate reduction significantly alters the action of the plant mitochondrial electron transport chain (mETC) [6]. In fact, the number of electrons consumed in nitrate reduction appears to be comparable to that in mETC [6,10]. In photosynthetic tissues, the expression and activities of several enzymes that bypass energy conservation steps in mETC are induced under limited nitrate reduction conditions, i.e., if the plant is experiencing N starvation or if ammonium is the sole N source [11,12,13,14,15]. One enzyme is the alternative oxidase (AOX), which allows direct electron transfer from ubiquinone to molecular oxygen, bypassing the proton-pumping complexes III and IV [16]. Therefore, upregulation of AOX under limited nitrate reduction may dissipate excessive reductants without being limited by steep proton gradients across the mitochondrial inner membrane, thereby attenuating reactive oxygen/nitrogen species (ROS/RNS) production and oxidative stress [17,18]. Indeed, in *Arabidopsis* grown under low nitrate conditions, the shoot expression of antioxidant enzyme genes was induced by disrupting the major isoform *AOX1a* [14]. Antisense suppression of Arabidopsis *AOX1a* was found to dramatically decrease the reducing state of ascorbate as an antioxidant under ammonium but not under nitrate [8]. Moreover, in tobacco cell cultures subjected to N starvation, antisense suppression of *AOX1* caused carbohydrate accumulation [19]. This suggests that, under limited nitrate reduction, NADH oxidation via AOX instead of nitrate reductase (NR) could replenish NAD^+^ to drive the glycolysis. Meanwhile, Arabidopsis *AOX1a* deficiency enhanced foliar nitrate assimilation under nitrate-replete conditions, implying competition between AOX and nitrate reduction for reductants [20].

The above studies suggest that AOX is tightly linked to nitrate reduction. However, these studies manipulated nitrate reduction activity by transferring plants to growth conditions that included no/low nitrate or the use of ammonium as the sole N source. Reduced nitrate reduction may therefore be accompanied by N starvation or ammonium toxicity [21], causing plant growth suppression and/or the initiation of stress responses. Moreover, since nitrate acts as a signal to alter genome-wide gene expression [22], a decrease in the nitrate supply also depletes the nitrate signal. For these reasons, it is impossible to distinguish whether the above-mentioned effects of AOX deficiency are caused by reduced nitrate reduction, N starvation, ammonium toxicity, or the lack of nitrate signal. To solve this problem, we developed a novel culturing system to manipulate the degree of nitrate reduction and AOX activity without causing N starvation, ammonium toxicity, or lack of nitrate signal. Using this system, we examined genome-wide gene expression and plant growth to better understand the relationship between AOX and nitrate reduction. Our results clearly support that AOX alleviates mitochondrial oxidative stress and sustains plant growth under limited nitrate reduction.

## 2. Materials and Methods

### 2.1. Plant Materials and Growth Conditions

In this study, we used *A. thaliana*, accession Columbia (Col-0), as a control line and homozygous *aox1a-1* (SALK_084897 [23]), *aox1a-2* (SAIL_030_D08 [23]), *nr* (*nia1-1/chl3-5*, [22]), *aox1a-1 nr*, and *aox1a-2 nr* mutants. The homozygous multiple mutants of *aox1a-1 nr* and *aox1a-2 nr* were produced via crossing. The surface-sterilized seeds were placed in the dark at 4 °C for three days to break seed dormancy. For in vitro culturing, we first grew plants on a sterile cellophane sheet [24] placed on a solid medium containing 30 mL of half-strength Murashige and Skoog (1/2-MS) salts without N supplemented with 2.5 mM (NH_4_)_2_SO_4_, 0.1% (*w*/*v*) MES-KOH (pH 6.7), 0.5% (*w*/*v*) sucrose, and 0.25% (*w*/*v*) gellan gum (Fujifilm Wako, Osaka, Japan). Plants were grown horizontally at 22 °C for 18 days under a fixed photosynthetic photon flux density (PPFD) of 30 µmol m^−2^ s^−1^ (constant light). Plants on cellophane were then transferred to a medium containing 30 mL of 1/2-MS salts without nitrogen supplemented with 0.05% (*w*/*v*) MES-KOH (pH 5.7), 0.5% (*w*/*v*) sucrose, and 0.25% (*w*/*v*) gellan gum (Fujifilm). The resulting mixture was incubated at 22 °C for 24 h, also under a PPFD of 30 µmol m^−2^ s^−1^ (constant light). Finally, the plants on cellophane were transferred to a medium containing 30 mL of 1/2-MS salts supplemented with 0.05% (*w*/*v*) MES-KOH (pH 5.7), 0.5% (*w*/*v*) sucrose, and 0.25% (*w*/*v*) gellan gum (Fujifilm) and then incubated at 22 °C under a PPFD of 150–200 µmol m^−2^ s^−1^ (constant light). For soil cultivation, plants were grown for 14 or 24 days under a PPFD of 100–130 µmol m^−2^ s^−1^ (i.e., on a 16 h/8 h light/dark cycle) at 22–23 °C using an Arasystem 360 kit (Betatech BVBA, Gent, Belgium) in a 1:1 mixture of nutrient-rich soil (Supermix A; Sakata Seed, Kyoto, Japan) and vermiculite.

### 2.2. Determination of In Vitro Nitrate Reductase Activities

We assessed in vitro NR activity as per a previously published method [25] with slight modification. Frozen samples were ground in a multi-bead shocker (Yasui Kikai, Osaka, Japan) using zirconia beads. The resulting powder was then mixed with 10 vol. of extraction buffer (50 mM HEPES-KOH, pH 7.6, 1 mM EDTA, 7 mM cysteine) and the extracts were centrifuged at 20,400× *g* at 4 °C for 10 min. Next, a 30 μL aliquot of the supernatant was added to 90 μL of assay buffer (50 mM HEPES-KOH, pH 7.6, 133 μM NADH, 2 mM EDTA, 6.67 mM KNO_3_). After incubation at 30 °C for 2 and 17 min, 24 μL of 1% (*w*/*v*) sulfanilamide solution in 1 N HCl and 24 µL of 0.02% (*w*/*v*) N-(1-naphthyl)ethylenediamine dihydrochloride solution were added to 48 µL aliquots of reaction mixture. Finally, nitrite content was determined based on absorbance readings at 540 nm. In vitro NR activity was calculated by determining the nitrite amount produced over a 15 min period.

### 2.3. Determination of In Vivo Nitrate Reductase Activities

We submerged fresh shoots in 50 vols. of reaction buffer (i.e., 100 mM sodium phosphate buffer, pH 7.4, 10 mM KNO_3_, 4% (*v*/*v*) n-propanol), followed by vacuum infiltration. The reaction mixture was then incubated at 30 °C for 1 h in the dark. The amount of nitrite produced was then visualized by mixing the supernatant with 1% (*w*/*v*) sulfanilamide solution in 1 N HCl, and 0.02% (*w*/*v*) N-(1-naphthyl)ethylenediamine dihydrochloride solution in a 2:1:1 ratio.

### 2.4. Determination of Nitrate Concentration

Nitrate content was quantified via a previously published protocol [26] with slight modifications. Nitrate was extracted with 10 vol. of deionized water at 100 °C for 20 min. Next, 10 µL of the supernatant was mixed with 40 µL of 5% (*w*/*v*) salicylic acid in concentrated sulfuric acid, and the resulting mixture was then incubated at room temperature for 20 min. A mock treatment of 40 µL of concentrated sulfuric acid was also produced. Finally, 1 mL of 8% (*w*/*v*) NaOH solution was added to the mixture, and nitrate was determined based on absorbance at 410 nm.

### 2.5. Determination of Total Protein

Total protein was determined as previously described [25]. Frozen samples were homogenized with a multi-bead shocker (Yasui Kikai) using zirconia beads. Total proteins were then extracted with 10 vol. of sample buffer [2% (*w*/*v*) SDS, 62.5 mM Tris-HCl (pH 6.8), 10% (*v*/*v*) glycerol, and 0.0125% (*w*/*v*) bromophenol blue] and HaltTM protease inhibitor cocktail (ThermoFisher Scientific, Tokyo, Japan), followed by incubation at 95 °C for 5 min. Extracts were then centrifuged at 20,400× *g* at 8 °C for 10 min and 10 μL aliquots were suspended in 500 μL of deionized water. Next, 100 μL of 0.15% (*w*/*v*) sodium deoxycholate was added, and the mixture was incubated at room temperature for 10 min. Then, 100 μL of 72% (*v*/*v*) trichloroacetic acid was added, followed by incubation at room temperature for 15 min and centrifugation at 20,400× *g* for 10 min. Precipitates were then air-dried and suspended in 25 µL of deionized water. This suspension was used to determine total protein concentrations using Takara BCA Protein Assay Kits (TaKaRa, Kusatsu, Japan).

### 2.6. Immunodetection of AOX and ACTIN Proteins

An amount of 12.35 µL of total protein extract was mixed with 0.65 µL of 1 M DTT, followed by incubation at 95 °C for 3 min. Then, 5 µL of mixture (~0.43 mg fresh weight) was then subjected to SDS-PAGE using a 12% Mini-PROTEAN TGX Gel (Bio-Rad, Hercules, CA, USA) before transferring to a Trans-Blot Turbo Mini PVDF membrane using the Trans-Blot Turbo Transfer System (Bio-Rad). Membranes were incubated for 1 h in a blocking buffer (50 mM Tris-HCl, 150 mM NaCl, pH 7.6, 0.1% (*v*/*v*) Tween-20) containing 5% (*w*/*v*) ECL Prime Blocking Agent (Cytiva, Tokyo, Japan). Membranes were then incubated for 1 h with a 1/200 dilution of monoclonal antibodies against AOX (AS10 699, Agrisera, Vännäs, Sweden) or a 1/2500 dilution of polyclonal antibodies against ACTIN (AS13 2640, Agrisera). Finally, the antigen–antibody complex was detected using a 1/10,000 dilution of horseradish peroxidase conjugated with anti-mouse IgG sheep antibodies (Cytiva) for AOX or anti-rabbit IgG donkey antibodies (Cytiva) for ACTIN by ECL prime chemiluminescent detection (Cytiva). Proteins blotted on membranes were stained using TaKaRa CBB Protein Safe Stain (TaKaRa). The chemiluminescence and CBB staining were visualized using ImageQuant LAS 500 mini (Cytiva).

### 2.7. RNA Extraction

Frozen samples were ground in a multi-bead shocker (Yasui Kikai) using zirconia beads. Next, total RNA was extracted using an RNeasy Plant Mini Kit (Qiagen, Tokyo, Japan) according to the manufacturer’s instruction. For RNA-seq library preparation, RNA was purified using on-column DNase digestion (Qiagen).

### 2.8. RNA-Seq

RNA quality was evaluated using a Qubit RNA IQ assay kit (ThermoFisher Scientific). RNA samples with RNA IQs 8.7–10.0 were used for library preparation. cDNA libraries were constructed using a NEBNext Ultra II RNA Library Prep Kit with sample purification beads (New England Biolabs, Tokyo, Japan), a NEBNext Poly(A) mRNA Magnetic Isolation Module (New England Biolabs), and a NEBNext Multiplex Oligos for Illumina (New England Biolabs). cDNA libraries were then sequenced using a NextSeq 500 (Illumina, Tokyo, Japan), and resulting bcl files were converted to fastq files using bcl2fastq (Illumina). The RNA-seq raw data are available in the ArrayExpress database under accession number E-MTAB-14027. The reads were analyzed according to the method described by Notaguchi et al. [27] and mapped to the Arabidopsis reference (TAIR10) using Bowtie [28] with the following options: “–all–best–strata”. Finally, obtained reads were analyzed using iDEP version 0.96 [29], Metascape [30], and GeneCloud [31] using the default settings.

### 2.9. RT-qPCR

Reverse transcription (RT) was performed using a ReverTraAce qPCR RT Master Mix with gDNA Remover (Toyobo, Osaka Japan). Synthesized cDNA was then diluted tenfold with water and used for quantitative PCR (qPCR). RT-qPCR was performed by a QuantStudio 1 (ThermoFisher Scientific) with KOD SYBR qPCR Mix (Toyobo). Relative transcript levels were calculated using the comparative cycle threshold method with *ACTIN3* as an internal standard [21]. Primer sequences are shown in Appendix A.

### 2.10. Statistical Analysis

The Tukey–Kramer multiple comparison test was conducted using R software v.2.15.3.

## 3. Results and Discussion

### 3.1. Manipulation of Nitrate Reduction and AOX Activities

We manipulated the activities of nitrate reduction and AOX via the transfer experiment (Figure 1A) using Col-0 and mutants deficient in either or both NR and AOX activities. First, plants were grown for 18 days with ammonium as the sole N source (Condition 1 in Figure 1A) to ensure uniform growth independent of NR activity. Higher medium pH and lower light intensity were used to reduce ammonium toxicity and AOX expression [11,13,21,32], allowing uniform growth regardless of AOX activity. Second, plants were subjected to N starvation at pH 5.7 for 24 h (Condition 2) to induce AOX expression [11,13]. Finally, plants were transferred to a medium containing adequate nitrate and incubated under moderate light intensity (Condition 3), which induced NR expression and filled photosynthetically-derived reductants to the cell [16,22,33]. The NR activity of Col-0 increased rapidly from 0 to 7 h after nitrate supply and slightly more from 7 to 24 h (Appendix A). Both in vitro and in vivo NR activities were increased 7 h after nitrate supply in the shoots of Col-0 and *aox1a-1* but not in *nr* and *aox1a-1 nr* (Figure 1B,C). Thus, we focused on the 7 h after nitrate supply to identify the early effects of nitrate reduction for subsequent experiments. We found that nitrate accumulated in the shoots of all lines after nitrate supply, with concentrations ranging from 12.1 to 22.4 µmol g^−1^ (Figure 1D), indicating an adequate supply of nitrate for signaling. Indeed, the expression of *NIA2* encoding the major NR isoform in leaves [34], which is inducible by nitrate signal [22,35], was strongly induced 7 h after nitrate supply in Col-0 and *aox1a-1* but not in *nr* and *aox1a-1 nr* (Appendix A). Higher nitrate concentrations in shoots of *nr* and *aox1a-1 nr* than Col-0 and *aox1a-1* would reflect a deficiency in nitrate reduction (Figure 1D). Shoot protein concentrations were comparable among all lines before and after nitrate supply, ranging from 27.0 to 29.7 mg g^−1^ (Figure 1E), suggesting that no N starvation occurred. RT-qPCR and Western blot analyses confirmed that the signals corresponding to *AOX1a* and AOX were negligible in *aox1a-1* and *aox1a-1 nr* (Appendix A), suggesting that the knockout of *AOX1a* was sufficient to diminish AOX, as reported earlier [13,14,23]. Taken together, these culturing conditions permit comparisons of plants’ differing nitrate reduction and AOX activity without causing ammonium toxicity, N starvation, and lack of nitrate signaling.

### 3.2. AOX1a Deficiency Induces Genes Related to Mitochondrial Oxidative Stress under Limited Nitrate Reduction

Next, to dissect the role of AOX in limited nitrate reduction, we performed three independent RNA-seq analyses using shoots from Col-0, *aox1a-1*, *aox1a-2*, *nr*, *aox1a-1 nr*, and *aox1a-2 nr* 7 h after nitrate supply (Appendix A). A k-means clustering analysis classified the transcripts into four groups according to their expression patterns (Figure 2A, Appendix A). In cluster D, *AOX1a* deficiency consistently induced gene expression in the *nr* background but not in the Col-0 background (Figure 2A,B). Moreover, 42 of 141 genes in cluster D were significantly upregulated in *aox1a-1 nr* and *aox1a-2 nr* relative to *nr* (Figure 2C and Appendix A). Further enrichment analyses of the 42 genes identified significant overrepresentation of the terms “toxin catabolic process”, “glutathione metabolism” (Figure 2D), “interpro-ipr004046 (glutathione S-transferase, C-terminal)”, and “gst” (Figure 2E). These terms were derived from the glutathione S-transferase (GST) genes *AT2G29460* (*GSTU4*), *AT1G17170* (*GSTU24*), *AT1G17180* (*GSTU25*), and *AT1G02920* (*GSTF7*) (Appendix A). GSTU4, a plant-specific tau class GST, may contribute to hydrogen peroxide degradation by using glutathione as an electron donor [36]. The terms “response to hypoxia” and “response to oxidative stress” were also enriched in these 42 genes (Figure 2D). The genes induced by hypoxia [37] and H_2_O_2_ treatments [38] were significantly upregulated in *nr*, and their induction was enhanced by *AOX1a* deficiency (Figure 2F,G and Appendix A). The NAC transcription factor ANAC017 mediates ROS-related mitochondrial retrograde signaling, thereby activating mitochondrial dysfunction stimulon genes including *AOX1a*, *UPOX*, and *ANAC013* [39,40]. The expression of ANAC017-inducible genes [39] and mitochondrial dysfunction stimulon genes [40] reached a maximum level in *aox1a-1 nr* and *aox1a-2 nr* (Figure 2H,I and Appendix A). Further RT-qPCR analyses revealed that, after nitrate supply, the expression of hypoxia-inducible genes (Figure 2J–M) and oxidative stress marker genes (Figure 2N–Q) was induced in *nr*, which was enhanced by *AOX1a* deficiency. Meanwhile, before nitrate supply, little difference was observed among all lines (Figure 2J–Q). It should be noted that hypoxia-inducible genes were induced in *aox1a-1 nr* and *aox1a-2 nr* (Figure 2F,J–M) despite no hypoxia treatment. Since hypoxia generally elevates cytosolic NADH/NAD^+^ ratio [41], the hypoxia-inducible genes may be upregulated by reductant accumulation. Indeed, *PDC1*, whose protein catalyzes the rate-limiting step of NADH-oxidizing alcohol fermentation in the cytosol [42,43], was dramatically induced in the shoots lacking both NR and AOX (Figure 2L), suggesting an accumulation of excessive reductants. Together, our transcriptome analysis suggests that AOX dissipates excessive reductants and mitigates oxidative stress under limited nitrate reduction.

### 3.3. AOX1a Deficiency Induces Genes for Respiratory Bypasses and Reductant Shuttles under Limited Nitrate Reduction

The plant mETC possesses type II NAD(P)H dehydrogenases located on the cytosol side (ND_ex_) or matrix side (ND_in_) of the inner mitochondrial membrane [6]. Since these dehydrogenases transfer electrons to ubiquinone and bypass the proton-pumping complex I, they can support the dissipation of excessive reductants. Of these, the expression of Arabidopsis *NDB2*, encoding the primary ND_ex_ contributor to NADH oxidation [44], was significantly induced in *nr* after nitrate supply, which was intensified by *AOX1a* deficiency (Figure 3A,B). Since cytosolic NR has a much lower K_m_ for NADH than ND_ex_ [11,44], NDB2 may operate only when nitrate reduction is limited. *NDA2*, encoding the NADH-oxidizing ND_in_ [6], also showed an expression pattern similar to *NDB2* after nitrate supply (Figure 3A,C). The compensated induction of *NDB2* and *NDA2* by *AOX1a* deficiency suggests a contribution of AOX1a-NDB2 and AOX1a-NDA2 modules to balance cellular redox under limited nitrate reduction. The co-function of AOX1a, NDB2, and NDA2 has already been suggested in plants subjected to various environmental stresses [44,45,46,47]. Also, the STRING database ver. 12.0 integrating protein–protein interactions [48] confirmed tight functional connections between AOX1a, NDB2, and NDA2 (Appendix A). Meanwhile, the expression of the uncoupling proteins genes (*UCPs*), whose proteins act as an uncoupler and/or as an aspartate/glutamate exchanger across the inner mitochondrial membrane [49], was little changed among the lines (Figure 3A).

The intracellular redox balance is tuned through the reductant shuttle systems across different cellular compartments [50,51,52,53,54]; electrons from membrane-impermeable NAD(P)H are temporarily stored in membrane-permeable compounds (e.g., malate, proline) through biochemical interconversions. The compounds are then transported across the membrane, followed by reconstitution of NAD(P)H. To reveal whether the redox perturbation due to deficiency of NR and/or AOX stimulates the shuttle systems, we surveyed transcriptional changes in the relevant genes (Figure 3A). *DiT1/OMT1*, which encodes the 2-oxogutarate/malate and oxaloacetate/malate translocator on the chloroplast inner membranes [51], was upregulated almost equally in *nr*, *aox1a-1 nr*, and *aox1a-2 nr* after nitrate supply (Figure 3A,D). In *A. thaliana*, the conversion of nitrite to ammonium by nitrite reductase occurs only in chloroplasts and requires six electrons (as six reduced ferredoxins) per nitrite. Hence, under limited nitrate reduction, DiT1/OMT1 as the malate valve [51,52,53] may export excessive reductants from chloroplasts, avoiding photo-oxidative stress. Meanwhile, *DIC3*, whose protein can transport malate from the mitochondria to the cytosol [52], was induced in *nr* after nitrate supply, and this induction was intensified by *AOX1a* deficiency (Figure 3A,E). The conversion of malate to oxaloacetate by cytosolic malate dehydrogenases is accompanied by the production of NADH [52,53]. A similar trend to *DIC3* was observed with *PRODH1* and *PRODH2* (Figure 3A,F,G), which encode proline dehydrogenases localized in mitochondria. Proline is produced in the cytosol by accepting electrons from NAD(P)H and can then be transported into the mitochondria [54]. PRODH1/2 catalyze the direct electron transfer from proline to ubiquinone via FAD as a cofactor in the mitochondria, bypassing complex I. Thus, in shoots lacking NR and AOX, the upregulation of *DIC3* and *PRODH1/2* would lead to dissipation of excessive reductants, implying over-reduction of mETC. These also support the hypothesis that AOX consumes excessive reductants under limited nitrate reduction.

### 3.4. AOX1a Deficiency Inhibits Shoot Growth under Limited Nitrate Reduction

Finally, we analyzed shoot growth parameters following nitrate supply. Over 6 days, we found that *AOX1a* deficiency significantly reduced (−15%) shoot fresh weight in the *nr* background but not in the Col-0 background (Figure 4A). Meanwhile, rosette diameter was little affected by *AOX1a* deficiency (Figure 4B). When plants were grown in nutrient-rich soil, shoot appearance and shoot fresh weight were decreased in *aox1a-1 nr* and *aox1a-2 nr* relative to the others (Figure 4C–E). Together, these suggest that AOX is crucial for sustaining plant growth under limited nitrate reduction.

## 4. Conclusions

We have successfully developed a cultural system to manipulate activities of nitrate reduction and AOX without causing ammonium toxicity, N starvation, or lack of nitrate signaling. Analyses using this system suggest that AOX alleviates mitochondrial oxidative stress and sustains plant growth under limited nitrate reduction. Our transcriptional dissection indicates redox/metabolic perturbation in plants lacking either or both NR and AOX. Further analysis of physiological and biochemical details using this system is awaited to better understand the relationship between nitrogen metabolism and respiration.

## Figures and Tables

**Figure 1 biomolecules-14-00989-f001:**
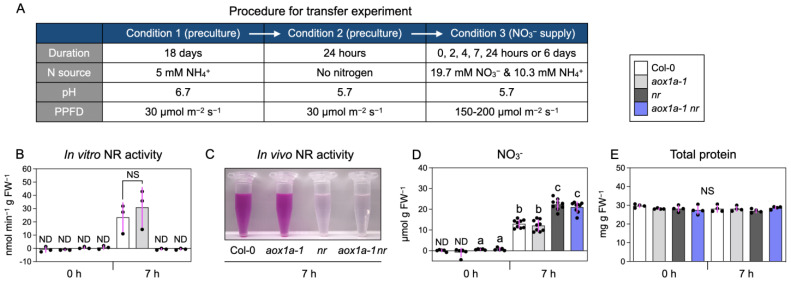
Manipulation of activities of nitrate reduction and AOX without causing N starvation, ammonium toxicity, or lack of nitrate signal. (**A**) Schematic diagram of experiment. (**B**) In vitro NR activity, (**C**) in vivo NR activity, (**D**) nitrate concentration, and (**E**) total protein concentration in plant shoots before and 7 h after nitrate supply. (**B**–**E**) Two plants of each line (eight in total) per plate were grown, and two shoots were pooled as one biological replicate. Data: mean ± SD (n = 3 (**B**), n = 5 (0 h) and 9 (7 h) (**D**), n = 4 (**E**)). For (**C**), color intensity is proportional to in vivo NR activity. Different lowercase letters indicate significant differences determined via Tukey–Kramer tests at *p* < 0.05. ND and NS denote “not detected” and “not significant”. FW denotes “fresh weight”.

**Figure 2 biomolecules-14-00989-f002:**
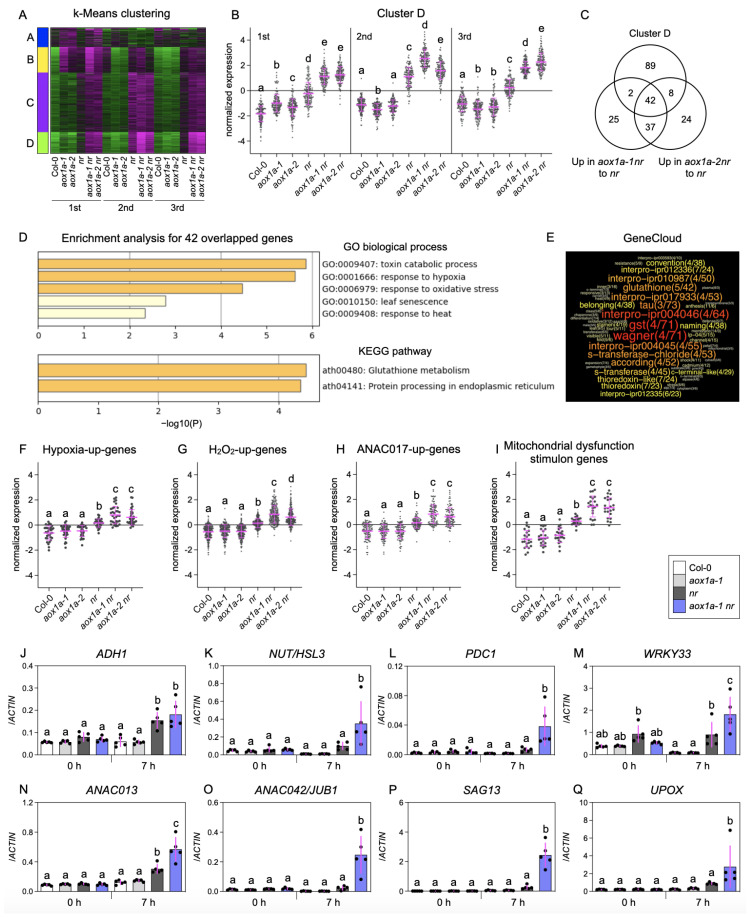
Transcriptomic alteration by *AOX1a* deficiency under limited nitrate reduction conditions. (**A**–**I**) Shoots from plants 7 h after nitrate supply were subjected to RNA-seq. One plant from each line (six in total) per plate was grown. Six shoots were pooled as one biological replicate. (**A**) Heat map from a k-means clustering of normalized transcript levels. Magenta and green represent higher and lower expression levels, respectively. Numbers (1st, 2nd, 3rd) indicate the order of independent RNA-seq experiments. (**B**) Plots of normalized transcript levels in cluster D. Normalized transcript levels of splice variants were averaged for each gene. (**C**) Venn diagram showing the number of genes from cluster D and genes significantly upregulated in *aox1a-1 nr* or *aox1a-2 nr* relative to *nr*. Enriched terms identified by Metascape (**D**) and GeneCloud (**E**) from the 42 overlapped genes in (**C**). (**E**) shows the number of genes containing the term (left) and the fold enrichment (right). Shoot expression of genes upregulated under hypoxia (**F**), by H_2_O_2_ (**G**), by ANAC017 (**H**), and included in mitochondrial dysfunction stimulon (**I**) 7h after nitrate supply. (**F**–**I**) Normalized transcript levels of splice variants were averaged for each gene using means from three independent experiments. RT-qPCR analysis of marker genes upregulated under hypoxia (**J**–**M**) and by oxidative stress (**N**–**Q**) in shoots before and 7 h after nitrate supply. (**J**–**Q**) Two plants of each line (eight in total) per plate were grown, and four shoots were pooled as one biological replicate. Data: mean ± SD (n = 5).

**Figure 3 biomolecules-14-00989-f003:**
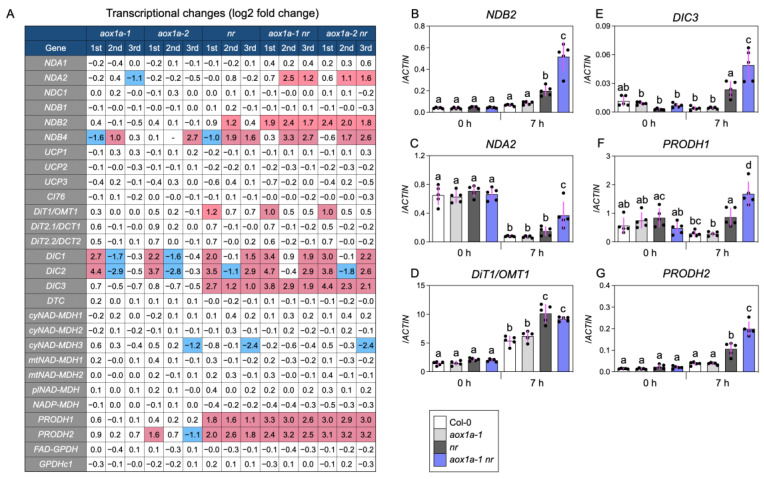
Effects of *AOX1a* deficiency on gene expression of type II NAD(P)H dehydrogenases, uncoupling proteins, and reductant shuttles under limited nitrate reduction conditions. (**A**) Transcriptional changes in genes encoding type II NAD(P)H dehydrogenases (*NDA1/2, NDC1*, *NDB1/2/4*), uncoupling proteins (*UCP1/2/3*), and complex I 76-kD subunit (*CI76*), and genes involved in reductant shuttles (*DiT1/OMT1*, *DiT2.1/DCT1*, *DiT2.2/DCT2*, *DIC1/2/3*, *DTC*, *cyNAD-MDH1/2/3*, *mtNAD-MDH1/2*, *plNAD-MDH*, *NADP-MDH*, *PRODH1/2*, *FAD-GPDH*, *GPDHc1*) in plant shoots 7h after nitrate supply. Transcriptional changes are represented as log2 fold change ratios against Col-0 based on the RPM (reads per million mapped reads) from the RNA-seq. Red and blue represent up- (>2-fold) and downregulation (<0.5 fold), respectively. Relative transcript levels of (**B**) *NDB2*, (**C**) *NDA2*, (**D**) *DiT1/OMT1*, (**E**) *DIC3*, (**F**) *PRODH1*, and (**G**) *PRODH2* in plant shoots before and 7 h after nitrate supply. Data: mean ± SD (n = 5).

**Figure 4 biomolecules-14-00989-f004:**
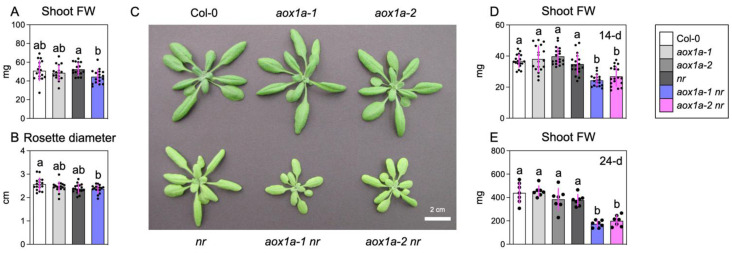
Effects of *AOX1a* deficiency on shoot growth under limited nitrate reduction conditions. (**A**) Shoot fresh weight and (**B**) rosette diameter of plants 6 days after nitrate supply. One plant of each line (four in total) per plate were grown and one shoot was regarded as one biological replicate. (**C**) Shoot appearance of 24-day-old plants and shoot fresh weights of (**D**) 14-day-old and (**E**) 24-day-old plants grown in pots containing nutrient-rich soil. Data: mean ± SD (n = 17 (**A**,**B**), n = 18 (**D**), n = 7 (**E**)).

## Data Availability

The RNA-seq raw data are available in ArrayExpress under accession number E-MTAB-14027 (https://www.ebi.ac.uk/biostudies/arrayexpress, accessed on 1 July 2024).

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
