# Peer review of "Alternative Oxidase Alleviates Mitochondrial Oxidative Stress during Limited Nitrate Reduction in Arabidopsis thaliana"

_biomolecules, 2024, doi:10.3390/biom14080989_

Round 1

Reviewer 1 Report

Comments and Suggestions for Authors

In the manuscript entitled:” "Alternative oxidase alleviates mitochondrial oxidative stress during limited nitrate reduction in Arabidopsis thaliana” by Otomaru D. et al., the authors describe the role of alternative oxidase in reducing oxidative stress in the presence of reduced nitrate reduction.

The authors investigated the changes in gene expression in a plant model in which the alternative oxidase and nitrate reductase were mutated. The double mutants were used to solve the problem of the link between AOX deficiency and nitrate reduction.

The novelty of this study is that a novel culture system was developed to manipulate the degree of nitrate reduction and AOX activity without causing N deficiency or ammonium toxicity. Shoot growth of A. thaliana in several AOX and nr mutants (aox1a-1 nr and aox1a-2 nr) is quite convincing regarding the role of AOX in maintaining plant growth under limited nitrate reduction.

Gene expression analysis confirms that AOX deficiency enhances the expression of genes encoding antioxidant enzymes and enzymes involved in excessive mitochondrial redox potential reduction, such as type II NAD(P)H dehydrogenases.

The article is well written and the illustrations are very clear

As the authors state, further biochemical analysis of mitochondrial functionality is required to confirm the role of AOX in mitochondrial oxygen consumption rate during reduced nitrate reduction.

Reviewer 2 Report

Comments and Suggestions for Authors

This article describes the roles of the alternative oxidase (AOX) and nitrate reductases (NRs) in Arabidopsis thaliana, with particular emphasis on the role of AOX on dissipation of reactive oxygen species (ROS) under conditions of low or no nitrate reduction. Importantly, the paper describes design of an experimental system that avoids causing N starvation, ammonium toxicity, and lack of nitrate signal, factors that would normally confound direct measurements of the critical components of this process. This method and comparative analysis of both aox mutants and aox nr double mutants allowed direct analysis of the roles of AOX when nitrate reduction was limited and the importance of this alternative component of the mitochondrial electron transport chain in stress tolerance. The evidence supporting these conclusions stems from in vitro and in vivo measures of nitrate presence, as well as transcriptomic comparisons between wild type and the various mutants and plant growth comparisons as well.  

The article is clearly, concisely, and completely written; the experiments are well-done and appropriate, providing compelling evidence for their conclusions. Very few typos or stylistic changes were identified.

Reviewer 3 Report

Comments and Suggestions for Authors

Comments on the Quality of English Language

Reviewer 4 Report

Comments and Suggestions for Authors

Why is section 2.1 in italics?

Fig 1B: why is 'n' a range from 5 to 9? Explain this in main text.

3.3 - why italicised?

Fig 3 - remove dots from columns, they are distracting.

Fig 4 - the dots on column bars are distracting - remove.

3.4 - remove italics.

The major flaw of this paper is the lack of protein analysis - immunoblot (ie. westerns) of even a few of the major proteins shown to be 'upregulated' or 'downregulated' via PCR. Include this. 
